# Clinical and molecular characteristics of COVID-19 patients with persistent SARS-CoV-2 infection

Bin Yang [1,2,4], Junpeng Fan [1,2,4], Jia Huang[1,2,4], Ensong Guo [1,2], Yu Fu[1,2], Si Liu[1,2], Rourou Xiao [1,2], Chen Liu[1,2], Funian Lu[1,2], Tianyu Qin[1,2], Chao He[1,2], Zizhuo Wang[1,2], Xu Qin[1,2], Dianxing Hu[1,2], Lixin You[1,2], Xi Li[1,2], Tian Wang[1,2], Peng Wu[1,2], Gang Chen[1,2], Jianfeng Zhou [3], Kezhen Li [1,2,5✉] & Chaoyang Sun [1,2,5✉]

The characteristics of COVID-19 patients with persistent SARS-CoV-2 infection are not yet well described. Here, we compare the clinical and molecular features of patients with long duration of viral shedding (LDs) with those from patients with short duration patients (SDs), and healthy donors (HDs). We find that several cytokines and chemokines, such as interleukin (IL)-2, tumor necrosis factor (TNF) and lymphotoxin α (LT-α) are present at lower levels in LDs than SDs. Single-cell RNA sequencing shows that natural killer (NK) cells and CD14$^+$ monocytes are reduced, while regulatory T cells are increased in LDs; moreover, T and NK cells in LDs are less activated than in SDs. Importantly, most cells in LDs show reduced expression of ribosomal protein (RP) genes and related pathways, with this inversed correlation between RP levels and infection duration further validated in 103 independent patients. Our results thus indicate that immunosuppression and low RP expression may be related to the persistence of the viral infection in COVID-19 patients.

[1] Department of Obstetrics and Gynecology, Tongji Hospital, Tongji Medical College, Huazhong University of Science and Technology, Wuhan, China. [2] Cancer Biology Research Center, Tongji Hospital, Tongji Medical College, Huazhong University of Science and Technology, Wuhan, China. [3] Department of Hematology, Tongji Hospital, Tongji Medical College, Huazhong University of Science and Technology, Wuhan, China. [4] These authors contributed equally: Bin Yang, Junpeng Fan, Jia Huang. [5] These authors jointly supervised this work: Kezhen Li, Chaoyang Sun. ✉email: tjkeke@126.com; suncydoctor@gmail.com

Currently, the world is witnessing a major devastating pandemic of coronavirus disease 2019 (COVID-19) caused by severe acute respiratory syndrome coronavirus-2 (SARS-CoV-2)[1]. On March 11th 2020, the World Health Organization has declared it a pandemic, which has had a profound impact on the global culture and economy[2]. Disease manifestation is highly heterogeneous, ranging from asymptomatic infection through mild to severe disease leading to death[3]. Moreover, the duration of viral shedding has been reported to vary dramatically, ranging from 6 to 105 days, with a median duration of 20 days from disease onset[4–7].

We previously reported that even in the 5th week after symptoms onset, the viral PCR positive rate of the tested samples remained around 20%[8]. More importantly, Victoria A et al.[7] reported that even 70 days after diagnosis, virus particles were observed in SARS-CoV-2 cultured in nasopharyngeal swabs though scanning and transmission electron microscopy, supporting persistent SARS-CoV-2 infection with shedding of infectious virus. Therefore, persistent infection potentially increases the risk of spread, resulting in the consumption of additional hospital resources and greater economic costs. Accordingly, in addition to focus on the severity of the COVID-19 disease, it is also important to explore the clinical, molecular characteristics and mechanism of long duration of viral infection. However, up to now, we have a limited understanding of the clinical and molecular characteristics of COVID-19 with long-term SARS-CoV-2 shedding.

The key point in SARS-CoV-2 persistent infection could be the depletion of antiviral defenses related to immune response[9]. The innate immune system is the first line of defense against infection from viruses[10]. While cytokines, such as interferon (IFN)-α/β/γ, TNF, and LT-α, have the potential to trigger the activation of intracellular antiviral pathways, other cytokines, such as interleukin (IL)-1α/β, IL-2, IL-6, and IL-12, are believed to indirectly promote antiviral responses by regulating various aspects of the immune response, including the autocrine or paracrine upregulation of IFN-α/β/γ and TNF[11]. On the contrary, accumulating evidences suggest that the overproduction of inflammatory cytokines response to SARS-CoV-2, so-called "cytokine storm", can cause organ damage, increase the mortality rate[12].

Furthermore, host adaptive immune responses are ultimately responsible for clearing respiratory viral infections. CD4+ and CD8+ T cells can perform various functions including: lysis of virally infected cells, production of cytokines to orchestrate the immune response, and initiation of a B cell response capable of producing antigen-specific IgG and IgA antibodies[13]. In contrast, regulatory T cells (Treg) inhibit the activation of both innate and adaptive immune cells via inhibitory surface molecules (like cytotoxic T-lymphocyte antigen-4 (CTLA-4) and lymphocyte-activation gene-3) and secretion of immunosuppressive cytokines (IL-10, transforming growth factor-β, and IL-35)[14]. Exploring the status and changes of adaptive immunity in long duration of viral shedding patients will give us a deeper understanding of the interaction of immune response and SARS-CoV-2.

In all biological cells, the task of synthesizing proteins is performed by complex molecular machines known as ribosomes. Ribosomes interact with messenger RNAs (mRNAs), which serve as blueprints for protein synthesis, and translate the nucleotide sequence of each mRNA into the amino-acid sequence of the corresponding protein[15]. Moreover, the production of ribosome is dysregulated by virus infection[16]. Decreased ribosome biogenesis may contribute to a poor immune response, including reduced IFN and antibodies production. Recently, several studies reported that nonstructural protein 1 (Nsp1) from SARS-CoV-2 efficiently interferes with 40S ribosome subunit, inhibits all cellular antiviral defense mechanisms, and then facilitates efficient viral replication and immune evasion[17,18]. However, whether ribosome biogenesis will impact the intrinsic immune responses of COVID-19 patients with persistent SARS-CoV-2 infection remains to be explored.

Herein, we examine the 48 serum cytokine/chemokine levels in 38 SDs, 12 LDs, and 22 HDs. Furthermore, fresh PBMCs from 3 HDs, 9 SDs and 5 LDs are collected and 10× Genomics single-cell RNA sequencing (scRNA-seq) is performed to dissect and compare the potential changes associated with persistent virus infection. We find that several cytokines/chemokines (IL-2, TNF and LT-α) are present at lower levels in LDs than SDs. Single-cell RNA sequencing indicates immunosuppression status and low RP genes expression in LDs. Further, we provide a resource to reveal the clinical and molecular features of COVID-19 patients with persistent SARS-CoV-2 infection, which may have important implications for understanding and controlling COVID-19.

## Results

**Demographic characteristics**. The duration of virus shedding was defined as the interval from illness onset until successive negative detection of SARS-CoV-2 RNA, consistent with other studies of COVID-19[19,20]. As of April 30, 2020, a total of 12 non-critical COVID-19 in-patients exhibited long duration of viral shedding (>45 days). Given that the median SARS-CoV-2 viral shedding duration is ~3 weeks[21], we also collected 38 age- and gender-matched non-critical COVID-19 in-patients whose viral shedding durations were <21 days for comparison (Supplementary Table 1). All the patients were identified as laboratory-confirmed SARS-CoV-2 infected patients at Tongji Hospital, Wuhan, China. The median viral shedding duration was 57 days (range: 45–100 days) and 16 days (range: 3–21 days) in LDs and SDs, respectively (log-rank $p < 0.0001$) (Supplementary Table 1). The basic demographic information and clinical parameters comparison of these patients are detailed in Supplementary Table 1. Notably, there were no significant differences between LDs and SDs in comorbidities, complete blood counts (white blood counts, lymphocyte counts, neutrophil counts, platelet counts, and hemoglobin), blood biochemistry (alanine/aspartate aminotransferase and lactate dehydrogenase), and coagulation function (prothrombin, activated partial thromboplastin time, and D-dimer). Moreover, inflammatory markers, such as procalcitonin, erythrocyte sedimentation rate (ESR), and c-reactive protein (CRP), which have been well reported as high-risk factors of the development of severe COVID-19[22–24], were also comparable in LDs and SDs. Therefore, further investigation is urgently needed to identify new indicators for viral shedding duration and the underlying mechanism of persistent viral shedding.

**Cytokines in SDs, LDs, and HDs**. Cytokines are central to the pathophysiology of COVID-19 and a "cytokine storm" has been described as a feature of COVID-19 severity, which is associated with adverse outcomes[25,26]. To further elucidate the immune response associated with the viral shedding duration, we examined the serum cytokine/chemokine levels in 38 SDs, 12 LDs, and 22 HDs. Intriguingly, among the 48 cytokines/chemokines detected, 21 inflammatory cytokines/chemokines had the lowest levels in LDs when compared to SDs or HDs (Fig. 1c). Of these, platelet-derived growth factor (PDGF-BB) ($p = 0.000065$), C-C motif ligand 5 (CCL5) ($p = 0.00011$), and macrophage migration inhibitory factor (MIF) ($p = 0.00015$) showed the most significant changes (Fig. 1c and Supplementary Fig. 1). In addition, IL-1β, IL-2, IL-2R, IL-9, IL-18, TNF, and LT-α, the upregulation of which contributed to lung injury, multiorgan failure, and ultimately death[27–29], were present at lower levels in LDs (Fig. 1c).

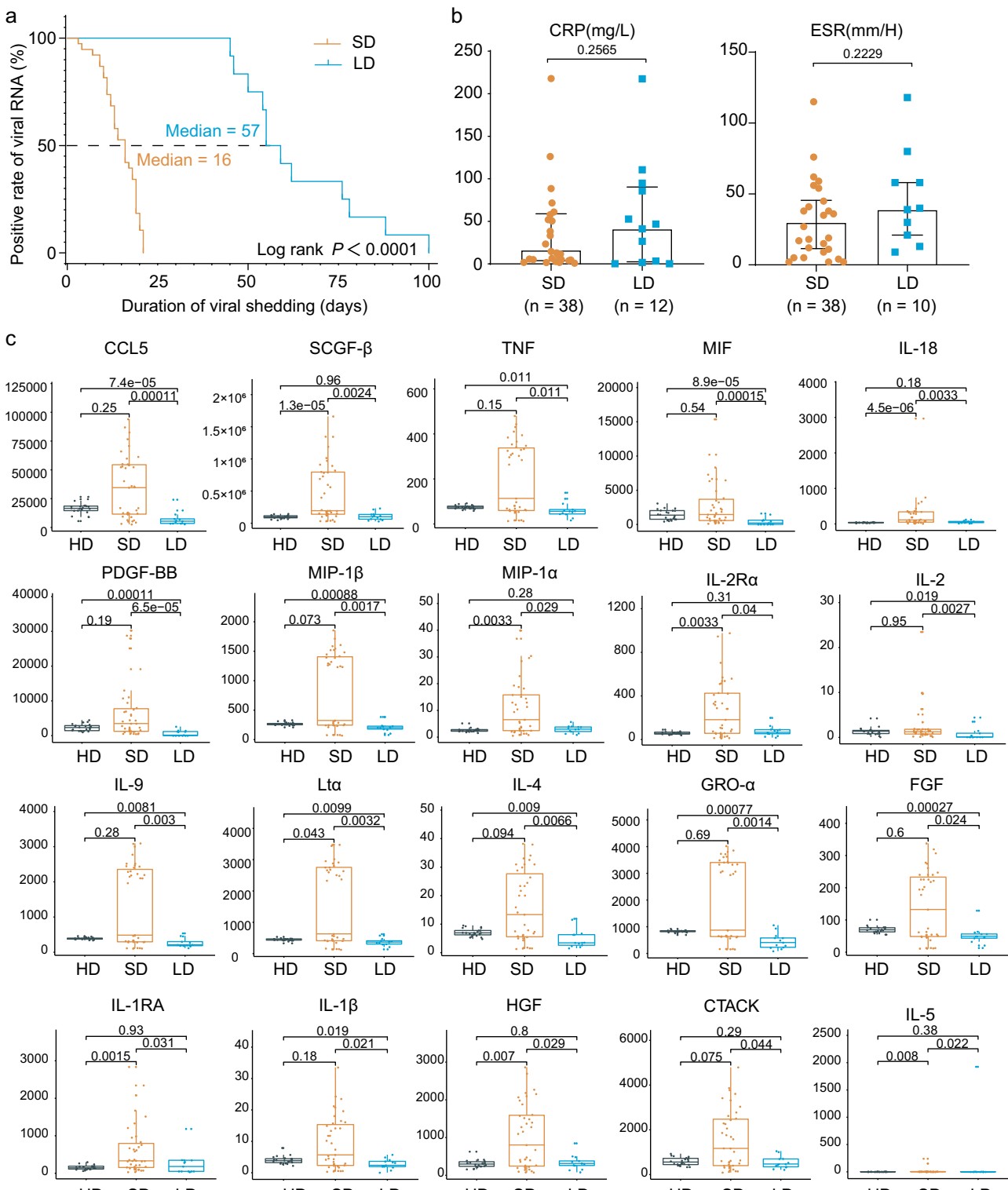

**Fig. 1 Virological, clinical and cytokines/chemokines characteristics in LDs and SDs. a** The Kaplan–Meier method was used to estimate the positive rate of viral RNA, and the two-sided log-rank test was applied to evaluate the significance difference of the duration of viral shedding in the LDs ($n = 12$) and SDs ($n = 38$). **b** The C-reactive protein (CRP) values obtained from clinical records of LDs ($n = 12$) and SDs ($n = 38$) and erythrocyte sedimentation rate (ESR) values obtained from clinical records of LDs ($n = 10$) and SDs ($n = 38$). The box plots show the median (middle line) and the first and third quartiles (boxes), whereas the whiskers show 1.5× the interquartile range (IQR) above and below the box. Non-paired two-tailed student $t$ test was performed. **c** Samples from HDs ($n = 22$), LDs ($n = 12$) and SDs ($n = 38$) were collected, and assays were performed to measure the concentrations of 48 cytokines/chemokines, of which 20 with statistically significant are shown. Y axis represents the cytokine concentration (pg/ml). The box plots show the median (middle line) and the first and third quartiles (boxes), whereas the whiskers show 1.5× the IQR above and below the box. One-sided Wilcoxon rank-sum test are performed. Source data are provided as a Source Data file.

Collectively, persistent viral shedding is associated with a weaker inflammatory response characterized by low circulating concentrations of cytokines and chemokines.

**Cell compositions differences of PBMCs in SDs, LDs, and HDs.**
To characterize the immunological features of LDs and SDs compared to HDs, we performed 10× Genomics scRNA-seq to study the transcriptomic profiles of PBMCs from 3 HDs, 9 SDs and 5 LDs (Supplementary Table 2). The demographics, clinical features, and laboratory findings of these patients are listed in Supplementary Table 2. After the unified single-cell analysis pipeline (see Methods), a total of 167,946 cells from all subjects (an average of 9879 cells per sample) were integrated into an unbatched and comparable dataset (Supplementary Table 2). In accordance with previous reports[30], we did not detect SARS-CoV-2 RNA expression in the PBMCs of these patients (Supplementary Fig. 2).

Using unsupervised clustering of uniform manifold approximation and projection (UMAP), we identified 20 cell populations based on the expression of canonical cell-type gene markers (Fig. 2a-b, Supplementary Fig. 3). To reveal the differences in cell composition across LDs and SDs and to compare them with that of HDs, we investigated the relative proportions of immune cells among the three groups (Fig. 2c-d). The proportion of NK cells in LDs was significantly reduced (Fig. 2d). CD14+ monocytes were lowest in LDs and exhibited a marginally significant decreasing trend in LDs when compared to SDs ($p = 0.06$). Notably, inflammatory monocytes, induced by T cells, have been reported to incite the cytokine storm in COVID-19[31]. The massive decreases in NK cells and CD14+ monocytes in LDs were in accordance with the observed decrease in inflammatory cytokines in LDs[32]. Moreover, the proportion of Treg was significantly highest in the LDs among three groups (Fig. 2d). Given the importance of Treg in secreting immunosuppressive cytokines and inhibiting the activation of both innate and adaptive immune cells[33,34], the statistically significant elevated levels of Treg may contribute to the suppression immune response observed in LDs. Taken together, the decreasing trend of NK cells and CD14+ monocytes, and the increased Treg may be associated with the immunosuppression status and the persistence of the virus in LDs.

**Transcriptional signatures associated with LDs.** Next, we performed hierarchical clustering based on relative gene expression changes with respect to the HDs to evaluate the molecular difference of each cell type in LDs and SDs. Unexpectedly, all cell types among the PBMCs clustered together according to the disease groups rather than by cell-types, with the exception of plasma B cells and megakaryocytes (Supplementary Fig. 4). This indicates that the molecular features of PBMCs in LDs and SDs are markedly different, regardless of the cell type. Therefore, we sought to identify variations in the relevant biological functions in individual cell types through differentially expressed genes (DEGs) and Gene Oncology (GO) analysis. Most importantly, we found that protein targeting to the membrane, endoplasmic reticulum (ER) related pathways, and translation related pathways were consistently downregulated in all cell types in LDs, with the exception of gamma delta (γδ) T cells, mucosal associated invariant T cells (MAIT), and megakaryocytes (Fig. 3a). In agreement with the GO analysis results, many genes encoding RP and immune related genes were specifically downregulated in LDs (Fig. 3b-c). Particularly, RPL41, RPS29, RPL36A, RPS27, RPS21, RPS10, RPL38, RPL39, and RPS28 localize to the ER and participate in protein synthesis, folding, and assembly, as detailed in the information provided on https://www.proteinatlas.org.

TMA7[35], TAF10[36], and PTOV1[37] were also specifically downregulated in LDs. These genes have previously been reported to be associated with ribosomes, and their overexpression promotes global protein synthesis. Given that antibodies[38] and cytokines[39] are synthesized, folded, modified, and assembled by the rough ER and attached ribosomes, these findings suggest that immune cells of LDs tend to have reduced cytokine synthesis, folding, and assembly functions, which is consistent with the lower levels of inflammatory cytokines observed in LDs (Fig. 1c). CEBPD and MAP2K2, which are involved in the production of proinflammatory cytokines, and RAC1[40], which is involved in IL-2 production, were selectively reduced in LDs (Fig. 3b-c). In addition, genes involved in T-cell activation (PCBP1, ARPC2), migration (FMNL1), cytotoxic function (GNLY, SRM), transcription factors (LYN), and downstream signal transduction (COTL1) were all reduced in LDs (Fig. 3b-c). Given that cytokines are produced by several immune cells, including adaptive T cells[41], the reduced cytokine levels in LDs are at least partially explained by these findings.

In addition, to better support the generalizability of the observations, we further reclassified, filtered published scRNA-seq data from a recent study[42]. Patients with fresh PBMC were included in the analysis if they met the following criteria: (1) HDs: Control group; (2) SDs: Days after symptom onset within 21 and is already during convalescence; (3) LDs: Symptoms are still developing more than 45 days after onset. Finally, the data of 38 COVID-19 patients (20 HDs, 16 SDs, and 2 LDs) were selected for analysis. Since the samples in LDs only have T-cell data (sorted by flow cytometry (CD3+)), we compared and analyzed T cells in this part of the data analysis. Using unsupervised clustering of UMAP, 9 cell populations based on the expression of canonical cell-type gene markers were identified (Supplementary Fig. 5a-c). In line with our results, GO analysis showed that in almost all T-cell subtypes in LDs, protein targeting to the membrane, ER related pathways, translation related pathways and immune response pathways were consistently downregulated (Supplementary Fig. 5d). These results collectively supported that cytokine synthesis, folding, and assembly functions in LDs may be reduced.

To further assess the association between ribosome proteins (RP) levels and the duration of virus shedding, we integrated bulk RNA-seq data from 103 independent COVID-19 patients. Remarkably, we found that lower expression of RPs was associated with longer viral shedding durations, including the following RPs identified in scRNA-seq data: RPL38, RPL41, and RPS10 (Fig. 3d-e). In summary, there is a negative correlation between RP levels and the duration of virus shedding. It is worthwhile to further explore whether specific RPs can be applied as indicators of persistent virus infection.

**Molecular features of T and NK cells in LDs and SDs.** We next performed sub-clustering analysis on T and NK cells considering their crucial antiviral effects[43,44]. UMAP embedding of T and NK cells from all the samples identified substantial differences in the cellular phenotypes of CD4+ T, CD8+ T, NKT, and NK cells (Fig. 4a-b). Also, the correlation matrix revealed that the molecular features differed between the two groups (Fig. 4c), such as memory CD8+ T cells and NK cells.

For example, in memory CD8+ T cells, DEGs involved in T-cell activation (SELENOK, FYN, CCL5, and RNF125), positive regulation of cytokine production (IRF1, SELENTOK, and HMGB2), proinflammatory mediators of secretion, and IFN-γ pathways (IRF1, HLA-DRB1, CCL5, and CCL4) were specifically downregulated in LDs, while they were upregulated in SDs compared to HDs (Fig. 5a-b).

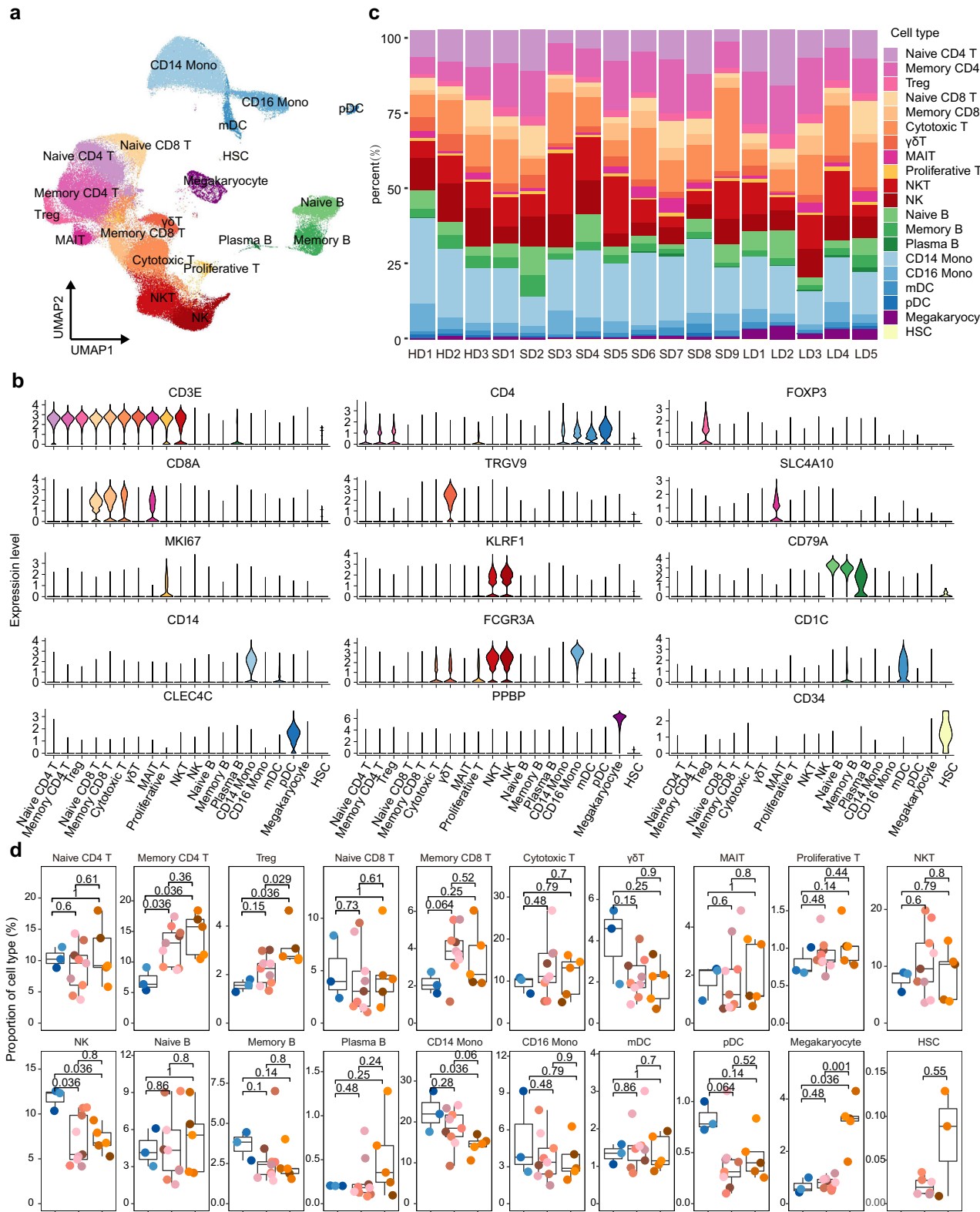

**Fig. 2 Differences in cell compositions by Single-cell transcriptomes of PBMCs. a** UMAP plot of 163,498 single cells colored by cell types identified from HDs ($n = 3$), SDs ($n = 9$), LDs ($n = 5$). **b** Expression distribution of selected canonical markers showed by violin plots in the 20 clusters. **c** Proportion of each cell type at single sample level. **d** Box plots of proportion of each cell cluster in each group. HDs ($n = 3$), SDs ($n = 9$), LDs ($n = 5$) were shown in different colors. Horizontal lines represent median values, with a maximum of 1.5 × interquartile range. One-sided Wilcoxon rank-sum tests were conducted between each group. Source data are provided as a Source Data file.

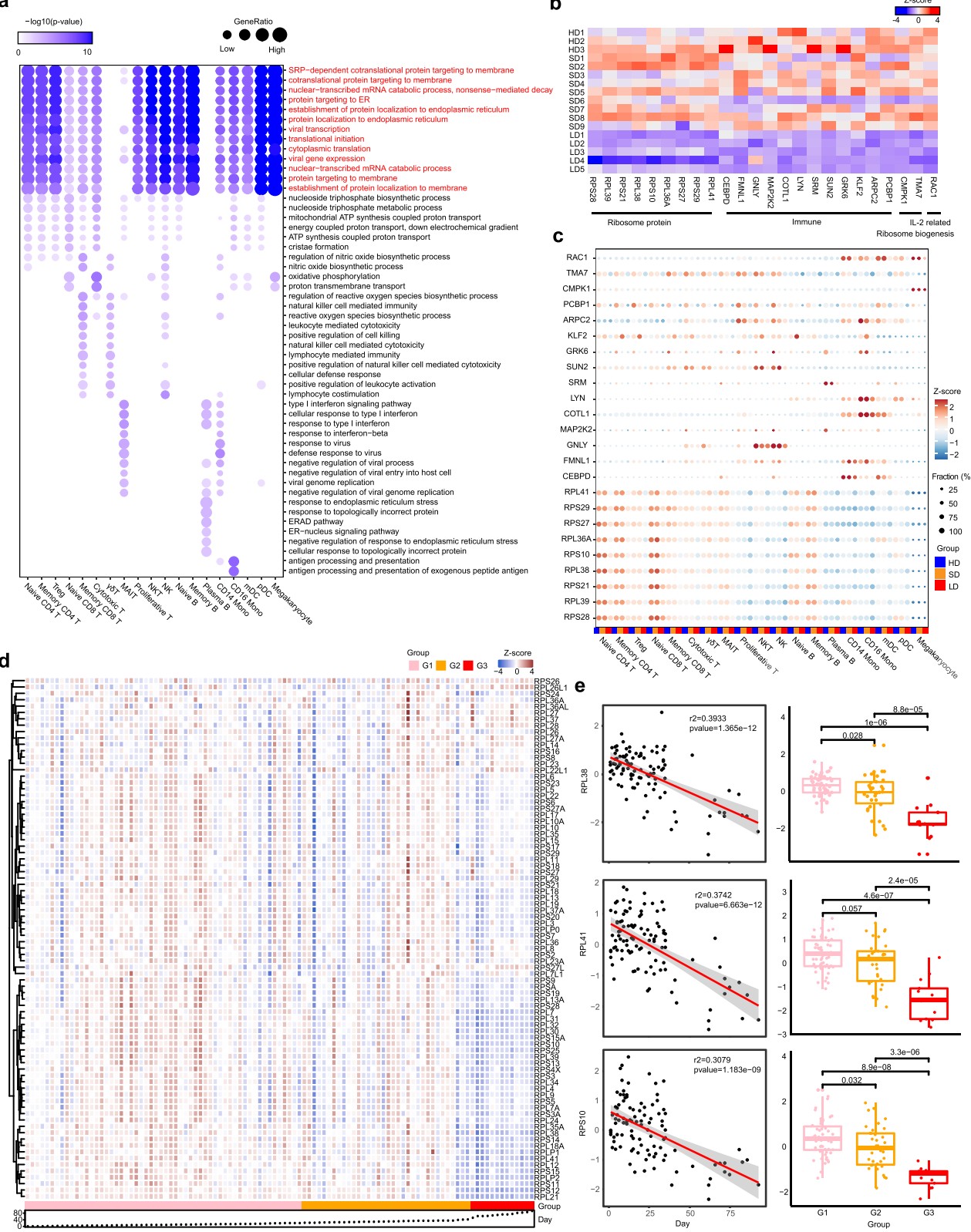

Given the importance of Treg in inhibiting the activation of both innate and adaptive immune cells[32,45]. Besides comparing the number of Treg (Fig. 2d), we further calculated their regulation score (see Methods) and the function score of Treg were unchanged between the three groups (Supplementary Fig. 6a). Conversely, CD8+ cytotoxic T lymphocytes play a key role in cell-mediated cytotoxicity against virus-infected target cells[46]. Another possibility was raised that immunosuppression LDs may have more exhausted cells. Therefore, we have developed exhaustion score to evaluate the exhaustion of cytotoxic T cell in three groups (see Methods). Notably, cytotoxic T cells in LDs showed significantly highest exhausted scores (Supplementary Fig. 6b). Once again, elevated counts of Treg and exhausted cytotoxic T cells may associate with

**Fig. 3 Transcriptional signatures associated with long viral shedding duration. a** Enriched GO pathways of downregulated genes between LDs and HDs at cell type resolution. Pathways enriched by ribosomal genes are labeled in red. The color intensity indicates the enrichment *p* values and the point size indicates the ratio of gene enrich in each pathway. **b** Expression level of selected genes across 17 samples. The color intensity indicates the relative expression level. **c** Expression of selected genes among groups at cell type resolution. The color intensity indicates the relative expression level and the point size indicates the ratio of cells with each gene expressed. The color bar under the dot plot indicates the group. **d** Expression level of ribosomal genes in whole blood bulk-RNA across 103 COVID-19 patients. The color intensity indicates the relative expression level, the color bar under the heatmap indicates the disease group and the scatter indicates the duration time of COVID-19. **e** Left: Correlation (with 95% confidence intervals) between expression levels of selected ribosomal genes and duration time of COVID-19 among 103 COVID-19 patients. Right: Boxplot of expression levels of selected genes in three groups among 103 COVID-19 patients (G1: *n* = 52; G2: *n* = 38; G3: *n* = 13). Horizontal lines represent median values, with a maximum of 1.5 × interquartile range. Difference between three groups were performed by one-sided Wilcoxon rank-sum tests. Source data are provided as a Source Data file.

immunosuppression status and persistence of viral shedding in LDs.

Next, we reconstructed T cell antigen receptor (TCR) sequences from the TCR sequencing data. Briefly, more than 70% of the cells in all the subsets had matched TCR information, with the exception of the γδT, NK and NKT subsets (Fig. 5c). Compared to HDs, clonal expansion was obvious in patients with COVID-19, especially in those with SDs of viral shedding (Fig. 5d). Consistent with reduced T cell and NK cell immune activation shown above, the proportion of large clonal expansions (clonal size >30), primarily in cytotoxic cells in LDs was low (Fig. 5d).

To explore the preferential V and J combinations in SDs and LDs, we first analyzed and listed the V and J combinations most frequently observed in the TCRs in all samples (Fig. 5e). Among these combinations, relatively frequent pairings of the TCR in HDs were TRBV28::TRVJ2-7 and TRAV29/DV5::TRAJ20, while TRAV29/DV5::TRAJ49 and TRBV9::TRBJ1-3 were frequent in LDs, and TRAV17::TRAJ48 and TRBV15::TRBJ2-5 were frequent in SDs (Fig. 5e). The selective usage of V(D)J genes suggests that different immunodominant epitopes may drive the molecular composition of T-cell responses and may be associated with long- or short- term virus infection.

In addition, in NK cells, DEGs associated with the positive regulation of T-cell activation pathways (ZFP36L2, KLF2, IRF1, LYN, RAC1, JUNB and CXCR4) were also profoundly reduction in LDs compared with SDs (Fig. 6a-b). Altogether, these results implied reduced T cell and NK cell immune activation, supporting the immunosuppressive status in LDs.

**Features of B cell subsets in LDs.** Some T cells and cytokines prime B cells for maturation, which go on to become plasma cells and produce pathogen neutralizing antibodies[47]. We subclustered B cells into three subsets according to the expression and distribution of canonical B-cell markers (Supplementary Fig. 7a-b). Compared with HDs, plasma B cells were not significantly increased in SDs, which may be due to sampling during the convalescent period[48] (Supplementary Fig. 7a, Fig. 2d). In LDs, despite viral persistence, the proportion of plasma cells was also extremely low, which may indicate that LDs fail to produce sufficient neutralizing antibodies (Supplementary Fig. 7a, Fig. 2d). Previous studies[6,49] have suggested that antibodies produced by plasma cells in response to SARS-CoV-2 during initial exposure disappeared within a few weeks, but memory B cells persisted for much longer. Therefore, we compared the expression profiles of memory B cells in the three groups. Interestingly, the pathways involved in T-cell differentiation (CD83, ZFP36L2, and GPR183) and cell growth and activation (CD83, ZFP36L2, GPR183, and PELI1) were selectively enriched in SDs but not LDs, indicating that B and T cells in LDs might fail to synergize to clear the virus (Supplementary Fig. 7c-d). Moreover, RAC1 and PDE4B, which positively regulate the production of cytokines such as IL-2, and the pathways

involved in leukocyte chemotaxis (LYN, DUSP1, and RAC1) were exclusively enriched in SDs (Supplementary Fig. 7c-d).

## Discussion

The duration of viral shedding has been reported to vary dramatically and the longest period of viral PCR positive lasts more than 100 days[7]. Our previous study[8] showed that even in the 5th week after the onset of symptoms, the positive rate of viral PCR in the tested samples remained around 20%. More alarmingly, recent study[7] reported that virus particles were still observed in SARS-CoV-2 cultured in nasopharyngeal swabs though transmission electron microscopy even 70 days after diagnosis, supporting that SARS-CoV-2 has the ability to alive for a long time and persistently contagious. Therefore, persistent infection potentially increases the risk of transmission, leading to the consumption of additional hospital resources and greater economic costs. Accordingly, in addition to paying attention to the severity of the COVID-19 disease, it is also important to explore the clinical, molecular characteristics and mechanisms of long-term of viral infections.

In this report, we found that clinical indexes, including contemporaneous CRP, D-dimers, IL-6, IL-8, and ESR (Fig. 1b, Supplementary Fig. 1) failed to distinguish the patients with persistent viral shedding. Interestingly, Treg were particularly elevated in LDs (Fig. 2d), though function score were comparable between three groups. We also observed decreases in NK cells and CD14+ monocytes in LDs, which were in accordance with the observed decrease in inflammatory cytokines in LDs. Notably, cytotoxic T cells in LDs showed significantly highest exhausted scores. Meanwhile, GO analysis and DEGs implied reduced T cell and NK cell immune activation, leading to lower proportion of large clonal expansions, primarily in cytotoxic cells in LDs. These results collectively reveal the immunosuppression status of LDs.

The immunological mechanisms for control of SARS-CoV-2 infection have not yet been clearly elucidated. There is no doubt that insufficient activation of type I and type III IFNs is a key contributor to innate immune failure to control viral persistence. Moreover, decades of immunological mechanistic research have showed that an intact T cell-mediated adaptive immune response is essential for clearing and maintaining long-term suppression of viral infections. This is supported by a significantly increased risk of viral reactivation in patients whose adaptive immune system is suppressed[50,51]. In addition, Marie Helleberg et al.[52] reported that in a severe COVID-19 patient with T- and B-cells impairment, after discontinuation of antiviral drug (remdesivir), the fever recurred and abnormalities of blood tests worsened, which indicated that remdesivir suppressed viral replication but was unable to eradicate the infection in immunocompromised individuals. Coincidentally, our previous study[8] found a poor immune response in persistent viral infectious patients. Similarly, Victoria A et al.[7] also reported that immunocompromised individuals may shed infectious virus longer than previously recognized. All these

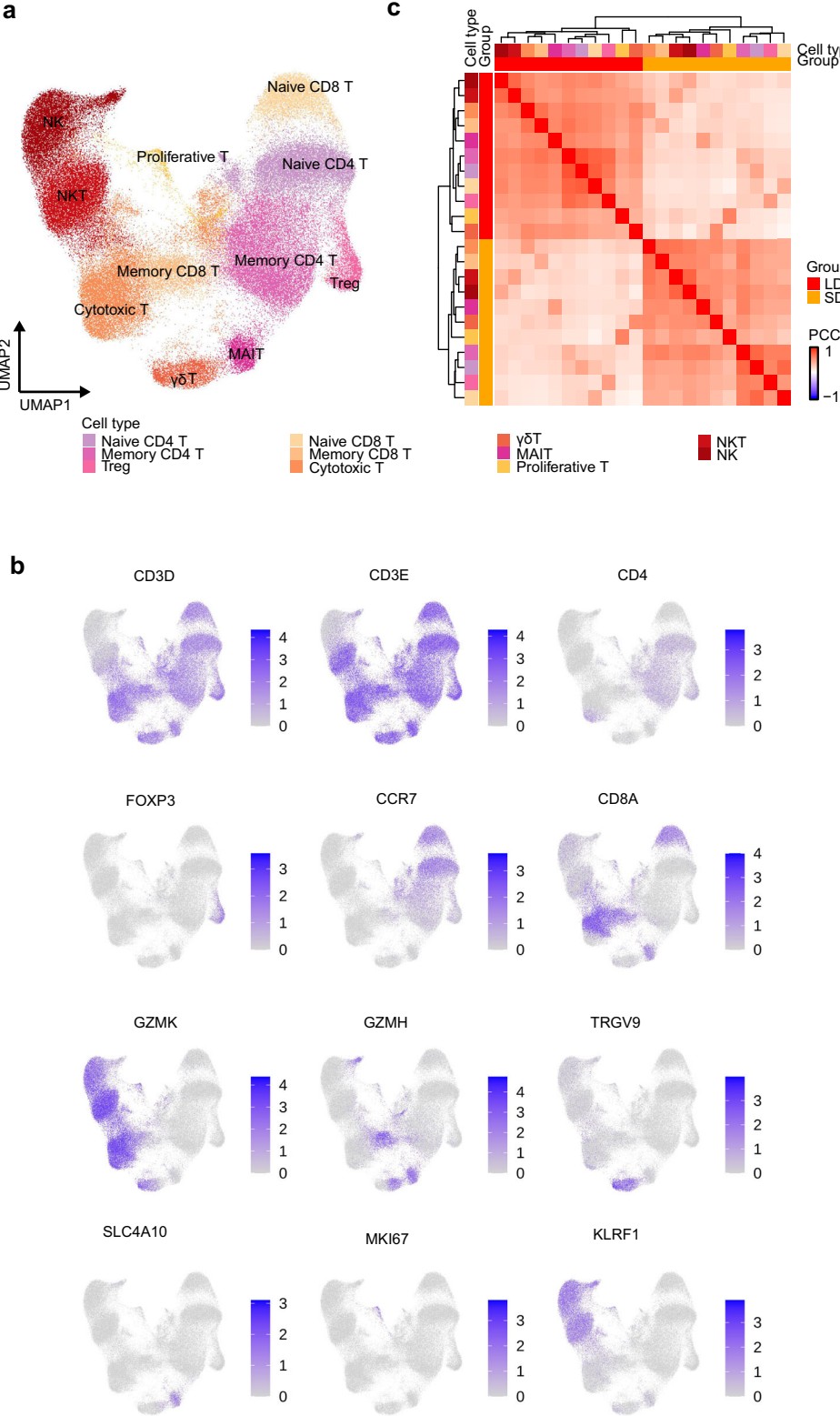

**Fig. 4 Sub-clustering analysis on T and NK cells. a** UMAP projection of T and NK cells. Each dot corresponds to a single cell, colored by cell type. **b** UMAP plot of canonical markers in 12 cell clusters. Data are colored according to log scaled expression levels. **c** Hierarchical clustering using the Pearson correlation coefficient (PCC) of a normalized transcriptome between disease groups in T and NK cells. The color intensity indicates the PCC and the color bars above the heatmap indicate the cell type and disease group.

results unanimously support our findings: immunosuppression may be related to the persistent viral infection.

In this study, the unique immunosuppression mediated by elevated Treg and exhausted cytotoxic T cells may responsible for

limiting excessive inflammation and play a vital role in preventing host tissues damage[53]. However, impaired viral clearance due to immunosuppression also cause prolonged contagious period after viral infection in LDs, increased risk of spread, consumption of

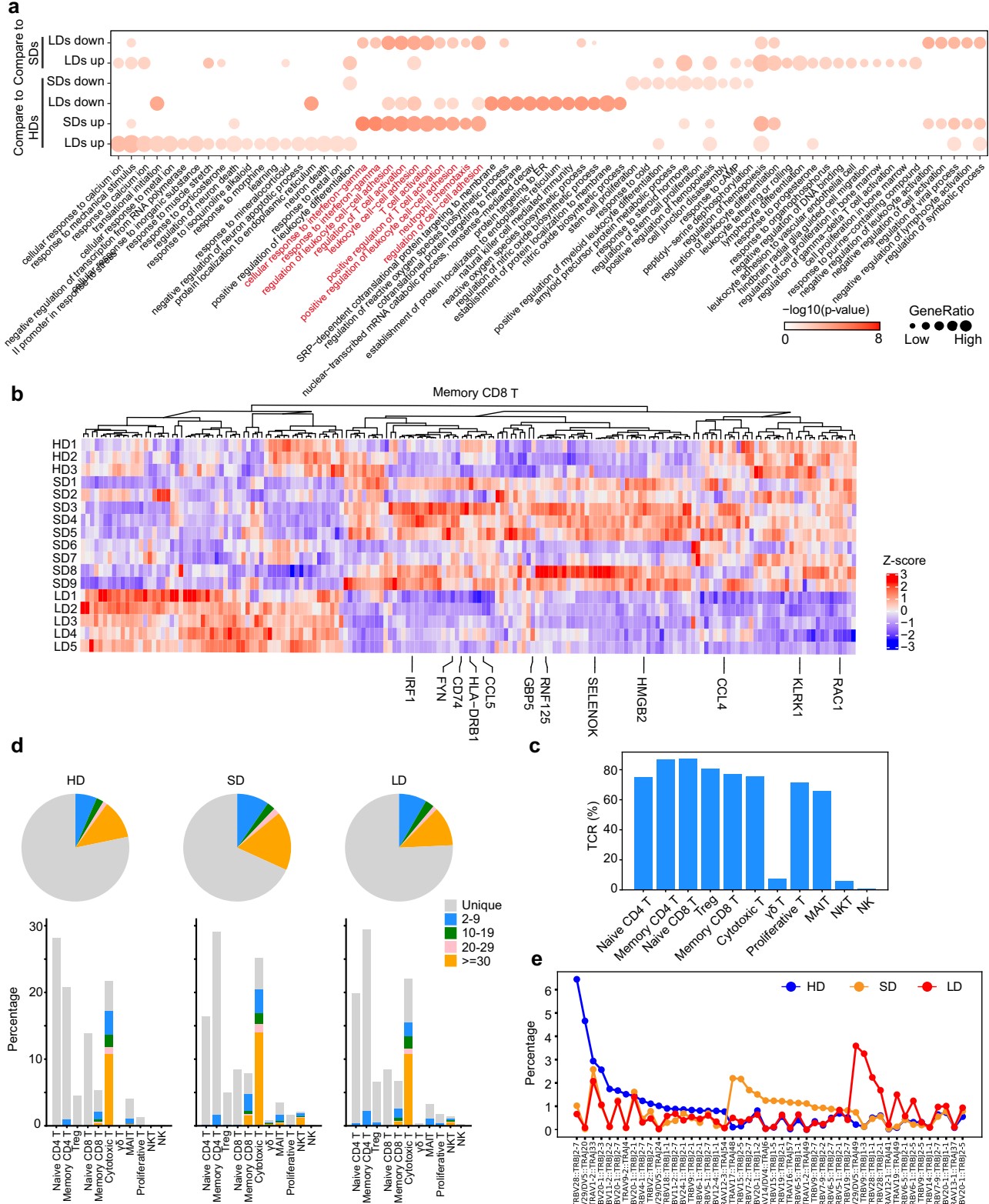

**Fig. 5 Molecular features of T cells in LDs and SDs. a** Enriched GO pathways of COVID-19 groups in memory CD8⁺ T cells (top 2 columns: DEGs between LDs and SDs, bottom 4 columns: DEGs for SDs and LDs compared to HDs). Pathway enriched by SDs upregulated genes are labeled in red. The color intensity indicates the enrichment *p* values and the point size indicates the ratio of gene enrich in each pathway. **b** Hierarchical clustering of expression of DEGs in memory CD8⁺ T cells at sample level. The color intensity indicates the relative expression of each gene. **c, d** The clonal status percentage of T cells and that at each cell type resolution in three groups. **e** The percentage of selected clonal types in three groups. Source data are provided as a Source Data file.

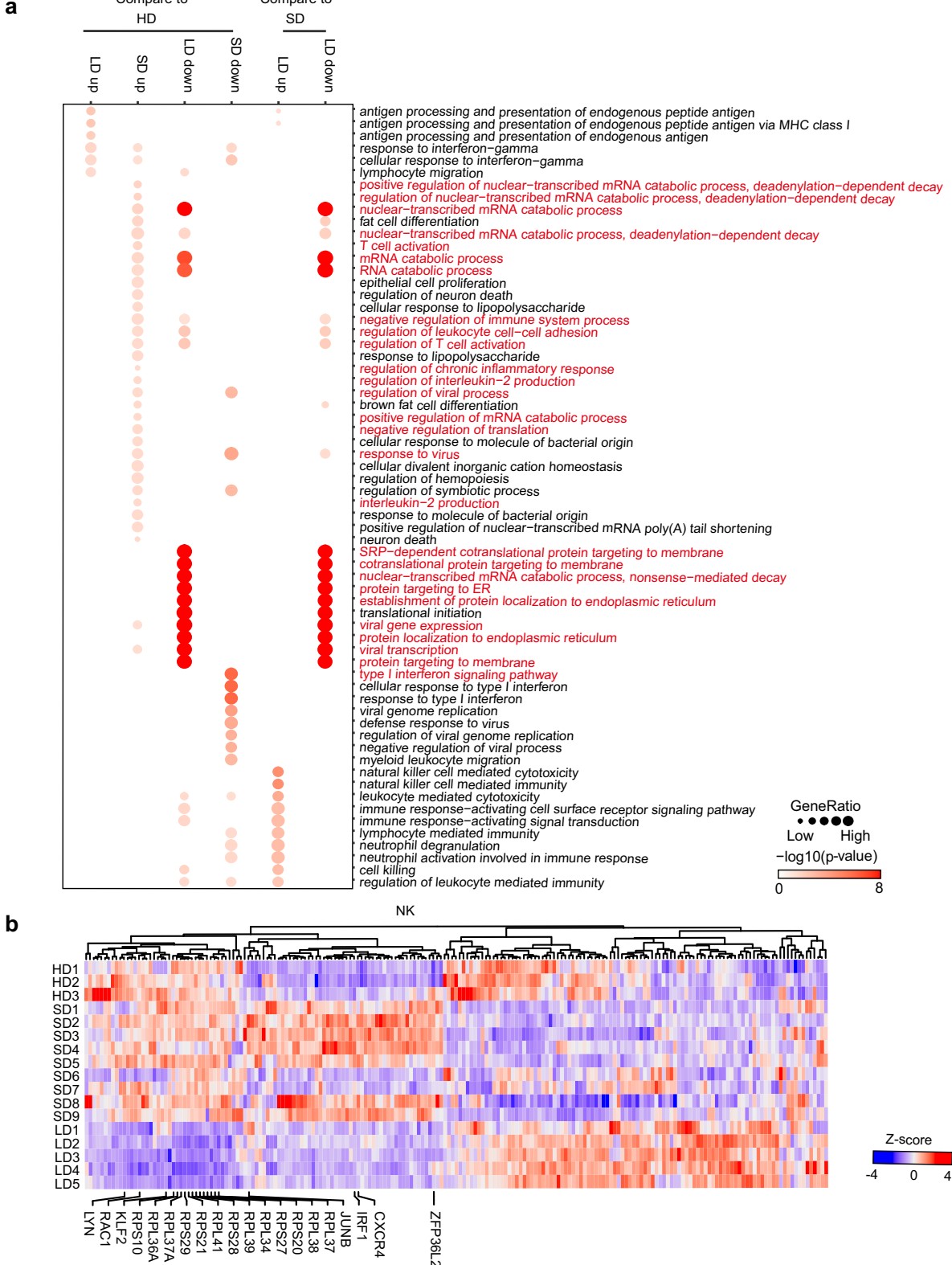

**Fig. 6 Immunological features of NK cells in LDs and SDs. a** Enriched GO pathways of COVID-19 groups in NK cells (top 2 columns: DEGs between LDs and SDs, bottom 4 columns: DEGs for SDs and LDs compared to HDs. Pathways enriched by SDs upregulated genes are labeled in red. The color intensity indicates the enrichment *p* values and the point size indicates the ratio of gene enrich in each pathway. **b** Hierarchical clustering of expression of DEGs in NK cells at sample level. The color intensity indicates the relative expression of each gene.

additional hospital resources, and greater economic costs. Accordingly, we need to develop different treatment strategies for persistent viral infections but uncritically ill patients. For these patients, it would be worth to further explore a combination of antiviral therapy (for example, remdesivir[54] or convalescent plasma[7]) and immune activation therapy (anti-PD-L1 or CTLA-4 antibody[55]).

Recently, several studies reported that Nsp1 from SARS-CoV-2 efficiently interferes with the cellular translation machinery (40S ribosome subunit), inhibits all cellular antiviral defense mechanisms, including the IFN response and other proinflammatory cytokines, and then facilitates efficient viral replication and immune evasion[17,18,56]. Indeed, we observed massively decreased RP expression in LDs in our study. However, whether the reduced levels of RPs in LDs are the cause or the consequence of viral persistence requires further investigation.

We used single-cell transcriptomics to characterize and visualize the peripheral immune responses in LDs compared to the responses in SDs and in HDs. We observed marked changes in the immune cell composition, molecular characteristic, and immunological features in LDs. Thus, this work provides new insights into the pathophysiology of COVID-19 and a resource for understanding peripheral immune heterogeneity in patients with long duration of viral shedding.

Lastly, there are a few limitations of our study. For example, our sample size is small, and the timing of the clinical presentation of the patients varied, which may influence their transcriptional landscapes. In addition, only peripheral blood was evaluated as it was challenging to obtain immune cells from the bronchoalveolar lavage fluid owing to biosafety concerns during the COVID-19 outbreak when this study was performed. Therefore, future studies with longitudinal samples from lesion sites, such as the lung, and with more patients can provide a more systematic overview and more comprehensive conclusions.

## Methods

**Patient information and data source**. This study was reviewed and approved by the Institutional Review Board of Tongji Hospital, Tongji Medical College, Huazhong University of Science and Technology (TJ-IRB20200405). All the enrolled patients signed an informed consent form, and all the blood samples were collected using the rest of the standard diagnostic tests, with no burden to the patients. A waiver of informed consent was obtained to query the patient electronic health records.

As April 30, 2020, serum from 50 patients and 22 healthy donors as controls were for the multiplex cytokine panel measurements. And PBMCs isolated from 14 patients and 3 healthy donors as controls were performed 10× scRNA-seq. Routine laboratory measurements and blood counts were obtained as part of standard medical care.

**Cytokine measurements**. The levels of serum cytokines were determined by Bio-Plex Pro Human Cytokines 48-Plex Screening assay (Bio-Rad Life Sciences, Hercules, CA, USA) using a Luminex FlEXMAP 3D system (Luminex, Austin, TX, USA) according to the manufacturer's protocols. The 48-Plex Screening panel: Basic FGF, CTACK, eotaxin, G-CSF, GM-CSF, GRO-α, HGF, ICAM-1, IFN-α2, IFN-γ, IL-1α, IL-1ra, IL-2, IL-2Rα, IL-3, IL-4, IL-5, IL-6, IL-7, IL-8, IL-9, IL-10, IL-12, IL-13, IL-15, IL-16, IL-17A, IL-18, IP-10, LIF, MCP-1, MCP-3, M-CSF, MIF, MIG, MIP-1α, MIP-1β, β-NGF, PDGF-BB, RANTES, SCF, SCGF-β, SDF-1α, TNF, LT-α, TRAIL, VCAM-1, VEGF-A. Data were analyzed using Bio-Plex Manager 6.2 software (Bio-Rad Life Sciences, Hercules, CA, USA). Undetected values were inputted with a random value between 0 and the limit of detection 1 to avoid an artificial reduction in the standard deviation.

**scRNA library construction sequencing by 10× genomics**. For both patients with COVID-19 and healthy controls, blood was collected into heparin tubes (Becton, Dickinson and Co.) and PBMCs were isolated by density gradient centrifugation using Ficoll-Paque Plus medium (GE Healthcare) and washed with Ca/Mg-free PBS. Blood was processed within 4 h of collection for all samples, and within 1 h for most. PBMC cells were examined by microscope after 0.4% Trypan blue coloring. When the viability of cells was higher than 80%, the experiment of library construction was performed using the ChromiumTM Controller and ChromiumTM Single Cell 5′ Reagent Version 2 Kit (10× Genomics, Pleasanton,

CA). Briefly, single cells, reagents and Gel Beads containing barcoded oligonucleotides were encapsulated into nanoliter-sized GEMs (Gel Bead in emulsion) using the GemCode Technology. Lysis and barcoded reverse transcription of polyadenylated mRNA from single cells were performed inside each GEM. Post RT-GEMs were cleaned up and cDNA were amplified. cDNA was fragmented and fragments end were repaired, as well A-tailing was added to the 5′ end. The adaptors were ligated to fragments which were double sided SPRI selected. Another double sided SPRI selecting was carried out after sample index PCR. The final library was quality and quantitated in two methods: check the distribution of the fragments size using the Agilent 2100 bioanalyzer, and quantify the library using real-time quantitative PCR (TaqMan Probe). The final products were sequenced using the MGISEQ-2000RS platform (BGIShenzhen, China).

**TCR V(D)J sequencing**. Full-length TCR V(D)J segments were enriched from amplified cDNA from 5′ libraries via PCR amplification using a Chromium Single-Cell V(D)J Enrichment kit according to the manufacturer's protocol (10× Genomics).

**Detection of SARS-CoV-2 transcripts**. Mock sample with SARS-CoV-2 transcripts was generated by add 200 SARS-CoV-2 paired reads to the health control. SARS-CoV-2 transcripts were identified from sequencing data using Viral-Track[57] and Cell Ranger (version 3.0.1, 10× Genomics) with a modified reference contain SARS-CoV-2 genome (NC_045512.2).

**Single cell RNA-seq data processing**. The sequencing data of patients were processed using Cell Ranger against the GRCh38 human reference genome. Quality of cells were then assessed based on the UMI counts per cell, genes expressed per cell and the proportion of mitochondrial gene counts using Seurat (version 4.0.0). Cells that had UMIs between 500 and 30,000, more than 200 genes expressed and fewer than 15% of UMIs from mitochondrial genes were considered high quality and retained for further analysis. We next identified and removed the doublets following previous described method[58]. After removing the doublets, a total of 167,946 cells were retained for downstream analysis.

To remove batch effect, the function "NormalizeData" and "FindVariableFeatures" in Seurat was performed respectively for each sample. After that, the dataset was scaled and PCA conducted with features exclude ribosomal protein and mitochondrial protein. Then, the dataset was integrated using Harmony[59] (version 1.0) and the cells were clustered using "FindNeighbors" and "FindClusters" function with parameter set to "resolution = 1". Finally, the cells were visualized by UMAP using the top 30 principal components.

**Cell types annotation**. The markers of each clusters identified by the "FindAll-Marker" function, as well as some canonical markers, were visualized using violin plot and feature plot, the expression of them were used to classify and annotate the clusters. HFSC cells and clusters expression more than 2 canonical cell type markers were excluded and 163,498 cells were retained for further analysis.

**DEGs analysis and GO enrichment**. DEGs were performed using "FindMarkers" function with MAST algorithm in Seurat based on a Bonferroni-adjusted $p < 0.05$ and a log fold change > 0.25. For GO enrichment, DEGs identified were conducted using function enrichGO in ClusterProfiler (version 3.1.8.1) with parameter set to "OrgDb = org.Hs.eg.db, ont = 'BP', pAdjustMethod = 'BH'".

**Hierarchical clustering of gene expression changes at cell type resolution**. Hierarchical clustering of gene expression changes was conducted following previous described method[60]. Briefly, the UMI count of each gene were normalized by the total UMI count in each cell type and multiplied by 100,000. The gene expression of each disease groups was divided by the values in the healthy donors and the highly variable genes in terms of the top 3000 standard deviation followed by log2-transformation. Hierarchical clustering was conducted based on the Pearson correlation coefficient (PCC) of the highly variable genes.

**Hierarchical clustering of ribosomal genes in whole blood RNA-seq**. Bulk RNA-seq data were downloaded from EMBL-EBI and accession codes is ERP127339. RNA-seq data of COVID-19 patients with duration time of viral shedding were enrolled in this study. Finally, 103 patients were divided into three groups separated by 21 days and 45 days were aligned to the reference genome using hisat2 and the gene expression level (FPKM) was calculated by RSEM. The expression of ribosomal genes was log2 transformed followed by z-transformation, which was used for hierarchical clustering.

**Analysis of T cells, NK cells and B cells**. T and NK cells were extracted from PBMCs and the cells were principal component analysis and visualized as described above. B cells in PBMCs were also extracted and processed using the procedure used for T cells.

**Defining cell regulation and exhaustion scores**. The regulation score in Treg based on the average expression of nine regulation-associated genes (CTLA4, PDCD1, TIGIT, LAG3, FOXP3, CCR7, LGALS3, TRAF1, IL2RA)[61]. The exhaustion scores were based on the average expression of six exhaustion-associated genes 6 exhaustion-associated genes (CTLA4, PDCD1, TIGIT, LAG3, HAVCR2, TOX)[62], the function "AddModuleScore" in Seurat was used to calculate the score with default parameters.

**Hierarchical clustering of DEGs**. DEGs of COVID-19 groups compared to HDs were selected and the expression of them in individuals are calculated by the function "Average Expression" in Seurat followed by z-transformation. Then the scaled expression was used for hierarchical clustering.

**TCR V(D)J analysis**. The sequencing data were performed using Cell Ranger V(D)J pipeline with GRCh38 as reference. The TCR matrix containing barcode information and clonotype frequency was obtained, the cells with at least one productive TRA and one productive TRB were retained for further analysis. Each unique TRA (s)-TRB(s) pair was defined as a clonotype.

**Analysis scRNA-seq data of COVID-19 patients from Zhang.et[42]**. scRNA-seq data of the study were downloaded from NCBI GEO database (GSE158055). Patients with fresh PBMC were included in the analysis if they met the following criteria: (1) HDs: Control group ($n = 20$); (2) SDs: Days after symptom onset within 21 and is already during convalescence ($n = 16$); (3) LDs: Symptoms are still developing more than 45 days after onset ($n = 2$). The data of selected samples was extracted and processed as described in our dataset. After annotation of clusters, the T cells were retained for downstream analysis.

**Boxplot**. All of the box plots in this paper were performed using "ggboxplot()" function in ggpubr R package. Each point represents for one sample. The horizontal line with each box represents the median, and the top and bottom of each box indicate the 25th and 75th percentile.

**Statistical analysis**. One-sided Wilcoxon rank-sum test were performed using R (version 4.0.2) in this study, *$p < 0.05$, **$p < 0.01$, ***$p < 0.001$. The two-sided log-rank test was performed using GraphPad Prism (version 8.0.2) in Fig. 1a. Non-paired two-tailed student $t$ test was performed using GraphPad Prism (version 8.0.2) in Fig. 1b. Non-paired two-tailed student $t$ test and Fisher's exact test was performed using SPSS (version 22.0) in supplementary table 1.

**Reporting summary**. Further information on research design is available in the Nature Research Reporting Summary linked to this article.

## Data availability

All data are available within the Article and Supplementary Files. The single-cell sequencing RNA data have been deposited to the European Bioinformatics Institute (EMBL-EBI,) and accession codes is ERP128255. We download and analysis scRNA-seq data from NCBI GEO database (https://www.ncbi.nlm.nih.gov/geo/) and accession codes is GSE158055. We also download and analysis bulk RNA-seq data of 103 COVID-19 patients from EMBL-EBI and accession codes is ERP127339. Source data are provided with this paper.

## Code availability

Computer code is available from GitHub under https://github.com/FlyPythons/Singlecell_COVID19_persistent_infection.

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

## Acknowledgements

We thank the anonymous referees for their useful suggestions and all the enrolled patients for their dedication to science.

## Author contributions

C.S. and K.L. conceived the experiments. B.Y. and J.H. collected the samples and performed single-cell library construction. J.F. and C.S. performed single-cell data analysis. E.G., Y.F., S.L., R.X., C.L., F.L., Z.W., X.L., T.Q., X.Q., D.H., L.Y., T.W., C.H., J.Z. collected clinical information, interpretation of data. B.Y., J.F. and J.H. wrote the paper and C.S., K.L., P.W., G.C., E.G., Y.F., S.L., R.X., X.W., C.L., F.L., Z.W., X.L., T.Q., X.Q., D.H., L.Y., T.W. C.H. and J.Z. revised it. P.W., G.C., K.L. and C.S. provided expertize and feedback. C.S. and K.L. conceived and coordinated the project.

## Competing interests

The authors declare no competing interests.
