## [Peer Review File · Nature Communications]

REVIEWER COMMENTS

Reviewer #1 (Viral immunity, cytokine response) (Remarks to the Author):

The authors describe the clinical and molecular characteristics of immune cell compartment, cytokines and ribosomal protein in noncritical patients with persistent SARS-Cov2 infection (LDs) using single cell RNA sequencing

They found that LDs patients show that amounts of most inflammatory cytokines/chemokines, such as IL-2, IL2R, TNF α / β , IL1 β , and CCL5 were present at low levels in the serum of LDs. Natural killer (NK) cells and CD14+ monocytes were decreased, while regulatory T cells (Tregs) were increased in the PBMC of LDs. Peripheral T and NK cells and memory B cells in LDs failed to activate, exhibiting reduced expression of ribosomal protein (RP) genes, indicating dysfunctions in cytokine/chemokine synthesis, folding, modification, and assembly.

These findings are informative for mechanism for persistence of viral shedding. My comments are as follows:

Comments

1. Studies have been conducted on a relatively small amounts of subjects: maybe too small to draw generalized conclusions.
2. Authors conclude that the negative correlation between the RP levels and viral shedding duration may help guide clinical management and resource allocation. However, I do not think that comparison of the RP levels is critical for determining course of severe COVID-19 disease between patients with COVID-19 with long durations (LDs) and short durations (SDs) of viral shedding because both are non-critical and neither LDc nor SDs may produce viable COVID-19 virus anymore for transmission.
3. Line 115: Authors speculate that increased Tregs in LDs may prevent the immune system from overreacting but contribute to the persistence of the virus. However,, anti-inflammatory cytokines such as IL10 are not significantly increased in the serum of LDs (extended Data Fig.1) Authors should discuss how Tregs contribute to dysfunction of the immune system in LDs, for example "CTLA-4 on Treg cells may contribute to prevent immune system".
4. Another possibility is raised that LDs may have more senescent/exhausted cells (PD1+,CD57+) in CD4+T cells and CD8+ T cells. Exhausted T cells show phenotypic features of an exhausted state including the upregulated expression of the inhibitory receptors programmed death (PD)-1. T cell exhaustion is characterized by functional unresponsiveness and prevent massive immunocivation in order to prevent tissue damage." Thus, the LDs patients display an exhausted T cell compartment, which may prevent immune system and cause persistence of viral shedding. Authors should add these data on exhausted T cell compartment in Fig.2
5. In the discussion section, authors should discuss more about significance for immunosuppression in patients with persistent infection. Elevated levels of serum proinflammatory cytokines and chemokines are known to contribute as "cytokine storm" to increased severity of disease caused by COVID-19. Unique immunoregulatory system mediated by Treg and (T cell exhaustion ?) are responsible for limiting excessive inflammation and play an important role in preventing lung and other organs from massive tissue damage.

Reviewer #2 (Transcriptome analysis, systems immunology) (Remarks to the Author):

In "Clinical and molecular characteristics of COVID-19 patients with persistent SARS-CoV-2 infection"

In this study the authors use single-cell transcriptomics and cytokine measurements to characterize immune responses in COVID19 patients. The authors analyzed samples from healthy donors, and non-critical patients with long- and short-term viral shedding. These analyses showed that the duration of viral shedding inversely correlated with immune activation of various types of immune cells and levels of pro-inflammatory molecules. A few hallmark immune cell populations were also reduced in the long shedding patients compared to healthy donors. An important finding is that there is a correlation between the low expression levels of genes encoding for the ribosomal machinery and the duration of viral shedding.

The sample collection, methods, and analysis are up-to-date and well-executed although some of the conclusion need to be revisited (see major concerns). Although, the patient's sample is small, as acknowledge by the authors, the study provides valuable information about COVID-19 infection and exemplify a new potential biomarker for diagnostics of long-viral shedders which can be used to design new and improved strategies, treatments and quarantine procedures. The study also uncovers new potential areas of research that address the issue of the ever increasing array of pathologies associated with COVID19 disease.

Major concerns:

The conclusions on the section " Differences in cell compositions detected using single-cell transcriptomes of human peripheral blood mononuclear cells" need to be revisited. Although there is a significant difference between the healthy donors and the long-durations group, there is no significant difference between the long- and short-duration group in CD4 T, T-reg, and NK cells. For example, the statistically significant elevated levels of T-regs in long-durations group compared to healthy donor is not an indication that this cell population plays a role in the suppression of cytokine production during COVID19 disease viral shedding duration because there is no statistically significant between the long- and short-duration groups even though they have significantly different cytokine profiles.

The analysis may be hindered by the possibility that within the short-durations group, 2 of the samples seems to be derived from the high cytokine producers and the other 2 from the high cytokine producers. This is suggested by the bimodal distribution of the cell counts in Fig 2d (Naive and memory CD4, T-reg, Naive B, etc) and the expression level on selected genes Fig 3b (S1 and S2 vs S3 and S4).

Minor concerns:

in line 31, LDs was defined before

in line 43, "LDs of viral shedding duration" is redundant as LDs is defined as Long Durations

in line 46, SDs was defined before

line 82 Please define IQR

In figure 1C, several cytokines and chemokines measured present a bimodal distribution (i.e. TNF- α , MIP-1 β , GRO- α , IL-9) within the short-duration group. It would be interesting to see if the scRNAseq analysis show differences in order single out potential drivers of the high or low production of

cytokines within this group. Do the low-cytokine-producers in the short-duration group have significant differential gene expression based on scRNAseq profiles to the long-durations group or to the healthy donors?

As in fig 1c, there seem to be a bimodal distribution in certain cell population shown in Fig. 2d.

Please soften this conclusion . "Taken together, these results demonstrated that the decreased NK cells and CD14+ monocytes, as well as increased Tregs in LDs may prevent the immune system from overreacting but contribute to the persistence of the virus."

line 222, GO pathways instead of go pathways?

line 252, there is no (extended data Fig. 4d-e). Suppl.Fig5?. If that is the case, there is no suppl. fig 5e

line 254, there is no (extended data Fig. 4c-d).

Responses to Reviewers

Reviewer #1:

1. Studies have been conducted on a relatively small amounts of subjects: maybe too small to draw generalized conclusions.

Response:

Appreciate for your valuable comment. In this revision, another five SDs samples were included. However, for LDs, although we tried our best, no more LDs samples were obtained because of the controlled epidemic of COVID-19 in China. Finally, scRNA-seq profiles of PBMCs samples of 3 HDs, 9 SDs and 5 LDs (a total of 167,946 cells) were thoroughly reanalyzed. Notably, our significant findings are particularly comparable with our previous insights (new Figure 2-4).

In addition, to better support the generalizability of the observations, we further reclassified, filtered published scRNA-seq data from a recent study¹. Patients with fresh PBMC were included in the analysis if they met the following criteria: 1) HDs: Control group; 2) SDs: Days after symptom onset within 21 and is already during convalescence; 3) LDs: Symptoms are still developing more than 45 days after onset. Finally, the data of 38 COVID-19 patients (20 HDs, 16 SDs, and 2 LDs) were selected for analysis. Since there are only T cell data in the two LDs, we compared and analyzed T cells among three groups. Importantly, in line with our research, GO analysis showed that in almost all T cell subtypes of LDs, protein targeting to the membrane, ER related pathways, translation related pathways and immune response pathways were consistently downregulated (Supplementary Fig.5).

Furthermore, to elucidate the associations between the inflammatory response and the viral shedding duration, we detected 48 serum cytokine/chemokine levels in 38 SDs, 12 LDs, and 22 HDs. And to further explore the association between the RP levels and the viral persistence, we integrated bulk-RNA-seq data from 103 independent COVID-19 patients. Taken together, the results above give us some reason to believe the conclusions were generalized.

2. Authors conclude that the negative correlation between the RP levels and viral shedding duration may help guide clinical management and resource allocation. However, I do not think that comparison of the RP levels is critical for determining course of severe COVID-19 disease between patients with COVID-19 with long durations (LDs) and short durations (SDs) of viral shedding because both are non-critical and neither LDs nor SDs may produce viable COVID-19 virus anymore for transmission.

Response:

Thanks so much for your constructive suggestion. The duration of viral shedding has been reported to vary dramatically. Our previous study² found that even in the fifth week after symptoms onset, the viral PCR positive rate of the tested samples remained around 20%. The longest period of viral PCR positive lasts more than 10 weeks. More importantly, Victoria A and colleagues³ reported that even 70 days after diagnosis, virus particles were observed in SARS-CoV-2 cultured in nasopharyngeal swabs through scanning and transmission electron microscopy, supporting persistent SARS-CoV-2 infection with shedding of infectious virus. Therefore, persistent infection potentially increases the risk of spread, resulting in the consumption of additional hospital resources and greater economic costs. Accordingly, in addition to focus on the severity of the COVID-19 disease,

it is also important to explore the clinical, molecular characteristics and mechanism of long duration of viral infection.

During our revision, several studies reported that nonstructural protein 1 (Nsp1) from SARS-CoV-2 efficiently interferes with the cellular translation machinery (40S ribosome subunit), inhibits all cellular antiviral defense mechanisms, including the interferon response and other proinflammatory cytokines, then facilitates efficient viral replication and immune evasion⁴⁻⁶, supporting our findings that RP level is related to the viral persistence.

In this revision, to avoid overinterpretation, we carefully changed the conclusion in Abstract to: "There is a negative correlation between RP levels and the duration of virus shedding".

3. Line 115: Authors speculate that increased Tregs in LDs may prevent the immune system from overreacting but contribute to the persistence of the virus. However, anti-inflammatory cytokines such as IL10 are not significantly increased in the serum of LDs (extended Data Fig.1) Authors should discuss how Tregs contribute to dysfunction of the immune system in LDs, for example "CTLA-4 on Treg cells may contribute to prevent immune system".

Response:

We appreciate your valuable suggestion. In this revision, after adding five SD patients we found the proportion of Treg were highest in the LDs among three groups with statistical significance (Fig.2d). We further calculated the regulation score in Treg based on the average expression of 9 regulation-associated genes (CTLA4, PDCD1, TIGIT, LAG3, FOXP3, CCR7, LGALS3, TRAF1, IL2RA)⁷, and the function score in Treg were unchanged (Supplementary Fig.6a). Taken together, elevated Tregs may be one of the factors that contribute to the suppression immune response and related to the persistent virus infection in LDs.

4. Another possibility is raised that LDs may have more senescent/exhausted cells (PD1+,CD57+) in CD4+T cells and CD8+ T cells. Exhausted T cells show phenotypic features of an exhausted state including the upregulated expression of the inhibitory receptors programmed death (PD)-1. T cell exhaustion is characterized by functional unresponsiveness and prevent massive immunoactivation in order to prevent tissue damage." Thus, the LDs patients display an exhausted T cell compartment, which may prevent immune system and cause persistence of viral shedding. Authors should add these data on exhausted T cell compartment in Fig.2

Response:

Thank you for the constructive suggestion. CD8+ cytotoxic T lymphocytes play a key role in cell-mediated cytotoxicity against virus-infected target cells⁸. Therefore, we calculated the exhaustion score in cytotoxic T cells, which were based on the average expression of 6 exhaustion-associated genes (CTLA4, PDCD1, TIGIT, LAG3, HAVCR2, TOX)⁹. Notably, cytotoxic T cells in LDs showed the highest exhaustion score (Supplementary Fig.6b), supporting that the LDs displayed exhausted cytotoxic T cells. Taken together, elevated counts of Treg and exhausted cytotoxic T cells may associated with immunosuppression status and persistence of viral shedding in LDs.

5. In the discussion section, authors should discuss more about significance for immunosuppression in patients with persistent infection. Elevated levels of serum proinflammatory cytokines and chemokines are known to contribute as "cytokine storm" to increased severity of disease caused by COVID-19. Unique immunoregulatory system mediated by Treg and (T cell exhaustion ?) are responsible for limiting excessive inflammation and play an important role in preventing lung and other organs from massive tissue damage.

Response:

Thank you so much for the constructive suggestion. We have added a discussion in the discussion section from line 309: “The immunological mechanisms for control of SARS-CoV-2 infection have not yet been clearly elucidated. There is no doubt that insufficient activation of type I and type III interferons is a key contributor to innate immune failure to control viral persistence. Moreover, decades of immunological mechanistic researches have showed that an intact T cell-mediated adaptive immune response is essential for clearing and maintaining long-term suppression of viral infections. This is supported by a significantly increased risk of viral reactivation in patients whose adaptive immune system is suppressed^{10,11}. In addition, Marie Helleberg et al¹² reported that in a severe COVID-19 patient with T- and B-cells impairment, after discontinuation of antiviral drug (remdesivir), the fever recurred and abnormalities of blood tests worsened, which indicated that remdesivir suppressed viral replication but was unable to eradicate the infection in immunocompromised individuals. Coincidentally, Our previous study² found a poor immune response in persistent viral infectious patients. Similarly, Victoria A and colleagues³ also reported that immunocompromised individuals may shed infectious virus longer than previously recognized. All these results unanimously support our findings: immunosuppression may be related to the persistent viral infection.

In this study, the unique immunosuppression mediated by elevated Treg and exhausted cytotoxic T cells may be responsible for limiting excessive inflammation and play a vital role in preventing host tissue damage¹³. However, impaired viral clearance due to immunosuppression also causes a prolonged contagious period after viral infection in LDs patients, increased risk of spread, consumption of additional hospital resources, and greater economic costs. Accordingly, we need to develop different treatment strategies for persistent viral infections but uncritically ill patients. For these patients, it would be worth to further explore a combination of antiviral therapy (for example, remdesivir¹⁴ or convalescent plasma³) and immune activation therapy (anti-PD-L1 or CTLA-4 antibody¹⁵).”

Reviewer #2:

1. The conclusions on the section "Differences in cell compositions detected using single-cell transcriptomes of human peripheral blood mononuclear cells" need to be revisited. Although there is a significant difference between the healthy donors and the long-durations group, there is no significant difference between the long- and short-duration group in CD4 T, T-reg, and NK cells. For example, the statistically significant elevated levels of Tregs in long-durations group compared to healthy donor is not an indication that this cell population plays a role in the suppression of cytokine production during COVID19 disease viral shedding duration because there is no statistically significant difference between the long- and short-duration groups even though they have significantly different cytokine profiles.

Response:

Thank you for the detailed comments. After five new SDs data were added in our analysis, the results are slightly different and the conclusions in line 97 have been revised to “The proportion of natural killer (NK) cells in LDs was significantly reduced (Fig.2d). CD14+ monocytes were lowest in LDs and exhibited a marginally significant decreasing trend in LDs when compared to SDs ($p=0.06$) (Fig.2d).” and in line 101 “Moreover, the proportion of regulatory T cells (Treg) was significantly highest in the LDs among three groups (Fig.2d).” Finally, we softened the conclusion in line 106 : “Taken together, the decreasing trend of NK cells and CD14+ monocytes, and the increased Treg may be associated with the immunosuppression status and the persistence of the virus in LDs.”

2. The analysis may be hindered by the possibility that within the short-durations group, 2 of the samples seems to be derived from the high cytokine producers and the other 2 from the high cytokine producers. This is suggested by the bimodal distribution of the cell counts in Fig 2d (Naive and memory CD4, T-reg, Naive B, etc) and the expression level on selected genes Fig 3b (S1 and S2 vs S3 and S4).

Response:

Thanks. After five new SDs data were added in our analysis, the bimodal distribution of the cell counts in SDs in Fig 2d was well corrected. Although there are individual variations of the expression level on selected genes in Fig 3b (SDs), it does not affect the conclusion.

In addition, to better support the generalizability of the observations, we further reclassified, filtered published scRNA seq data from a recent study¹. Patients with fresh PBMC were included in the analysis if they met the following criteria: 1) HDs: Control group; 2) SDs: Days after symptom onset within 21 and is already during convalescence; 3) LDs: Symptoms are still developing more than 45 days after onset. Finally, the data of 38 COVID-19 patients (20 HDs, 16 SDs, and 2 LDs) were selected for analysis. Importantly, in line with our study, GO analysis showed that in almost all T cell subtypes in LDs, protein targeting to the membrane, ER related pathways, and translation related pathways were consistently downregulated (Supplementary Fig.5).

3. in line 31, LDs was defined before

Response:

Thank you for reminding and we deleted this definition here.

4. in line 43, "LDs of viral shedding duration" is redundant as LDs is defined as Long Durations

Response:

Thank you for the detailed comment. We have corrected it accordingly.

5. in line 46, SDs was defined before

Response:

Thank you for reminding and we have deleted this definition here.

6. line 82 Please define IQR

Response:

As suggest, we have added the definition of IQR.

7. In figure 1C, several cytokines and chemokines measured present a bimodal distribution (i.e. TNF- α , MIP-1 β , GRO- α , IL-9) within the short-duration group. It would be interesting to see if the scRNAseq analysis show differences in order single out potential drivers of the high or low production of cytokines within this group. Do the low-cytokine-producers in the short-duration group have significant differential gene expression based on scRNAseq profiles to the long-durations group or to the healthy donors?

Response:

Thanks. It will be really interesting to see if the scRNAseq analysis show differences in order single out potential drivers of the high or low production of cytokines within SDs. According to your suggestion, we first carefully reviewed cytokines and chemokines results, and confirmed that these data were accurate. Unfortunately, the cytokine test cases and the subsequent single-cell sequencing cases were not from the same group of people. So, we have no data of cytokines and chemokines levels in SDs who were profiled by scRNAseq analysis. We totally appreciated your valuable suggestion.

8. As in fig 1c, there seem to be a bimodal distribution in certain cell population shown in Fig. 2d.

Response:

As request, 5 new SDs single-cell sequencing data were added in our analysis, the bimodal distribution of the cell counts in SDs was corrected.

9. Please soften this conclusion: "Taken together, these results demonstrated that the decreased NK cells and CD14+ monocytes, as well as increased Tregs in LDs may prevent the immune system from overreacting but contribute to the persistence of the virus."

Response:

Based on your suggestion, we softened the conclusion in line 106 : "Taken together, the decreasing trend of NK cells and CD14+ monocytes, and the increased Tregs may be associated with the immunosuppression status and the persistence of the virus in LDs."

10. line 222, GO pathways instead of go pathways?

Response:

Thank you for reminding and we have change "go" to "GO".

11. line 252, there is no (extended data Fig. 4d-e). Suppl.Fig5?. If that is the case, there is no suppl. fig 5e

Response:

Sorry for the mistake and we have corrected it in this revision.

12. line 254, there is no (extended data Fig. 4c-d)

Response:

Sorry for the mistake and we have corrected it in this revision.

1 Ren, X. *et al.* COVID-19 immune features revealed by a large-scale single-cell transcriptome atlas. *Cell* (2021).

2 Fu, Y. *et al.* Dynamics and Correlation Among Viral Positivity, Seroconversion, and Disease Severity in COVID-19: A Retrospective Study. *Annals of Internal Medicine* (2020).

3 Avanzato, V. A. *et al.* Case study: prolonged infectious SARS-CoV-2 shedding from an asymptomatic immunocompromised individual with cancer. *Cell* **183**, 1901-1912. e1909 (2020).

4 Schubert, K. *et al.* SARS-CoV-2 Nsp1 binds the ribosomal mRNA channel to inhibit translation. *Nature structural & molecular biology* **27**, 959-966 (2020).

- 5 Zhang, K. *et al.* Nsp1 protein of SARS-CoV-2 disrupts the mRNA export machinery to inhibit host gene expression. *Science advances* **7**, eabe7386 (2021).
- 6 Thoms, M. *et al.* Structural basis for translational shutdown and immune evasion by the Nsp1 protein of SARS-CoV-2. *Science* **369**, 1249-1255, doi:10.1126/science.abc8665 (2020).
- 7 Pfoertner, S. *et al.* Signatures of human regulatory T cells: an encounter with old friends and new players. *Genome biology* **7**, 1-18 (2006).
- 8 Andersen, M. H., Schrama, D., thor Straten, P. & Becker, J. C. Cytotoxic T cells. *Journal of Investigative Dermatology* **126**, 32-41 (2006).
- 9 Kim, K. *et al.* Single-cell transcriptome analysis reveals TOX as a promoting factor for T cell exhaustion and a predictor for anti-PD-1 responses in human cancer. *Genome medicine* **12**, 1-16 (2020).
- 10 Broers, A. E. *et al.* Increased transplant-related morbidity and mortality in CMV-seropositive patients despite highly effective prevention of CMV disease after allogeneic T-cell-depleted stem cell transplantation. *Blood, The Journal of the American Society of Hematology* **95**, 2240-2245 (2000).
- 11 Shah, K. V., Daniel, R. W., Zeigel, R. F. & Murphy, G. P. Search for BK and SV40 virus reactivation in renal transplant recipients. *Transplantation* **17**, 131-134 (1974).
- 12 Helleberg, M. *et al.* Persistent COVID-19 in an immunocompromised patient temporarily responsive to two courses of remdesivir therapy. *The Journal of infectious diseases* **222**, 1103-1107 (2020).
- 13 Kusnadi, A. *et al.* Severely ill COVID-19 patients display impaired exhaustion features in SARS-CoV-2-reactive CD8+ T cells. *Science immunology* **6** (2021).

- 14 Grein, J. *et al.* Compassionate use of remdesivir for patients with severe Covid-19. *New England Journal of Medicine* **382**, 2327-2336 (2020).
- 15 Pickles, O. J. *et al.* Immune checkpoint blockade: releasing the breaks or a protective barrier to COVID-19 severe acute respiratory syndrome? *British Journal of Cancer* **123**, 691-693 (2020).

REVIEWERS' COMMENTS

Reviewer #1 (Remarks to the Author):

The authors responded almost satisfactorily in line with the referee's comments and suggestions. I have only one minor comment as follows:

Line 126. "Important of Treg in secreting immunosuppressive cytokines (especially in IL-2 production (Supplementary Fig.1) " should be changed to "Important of Treg in secreting immunosuppressive cytokines (especially in IL-10 production (Supplementary Fig.1) ".

Reviewer #2 (Remarks to the Author):

The authors have addressed my comments. The paper is now suitable for publication. It is an important contribution.

Response to Reviewers

Dear Reviewers:

We are very much appreciated your constructive comments and recognition of our work. In this revision, we try to address the following issues as best as possible. Here, we give the following point-by-point responses to each reviewer and editor:

Reviewer #1:

1. The authors responded almost satisfactorily in line with the referee's comments and suggestions. I have only one minor comment as follows:

Line 126. "Important of Treg in secreting immunosuppressive cytokines (especially in IL-2 production (Supplementary Fig.1) "should be changed to "Important of Treg in secreting immunosuppressive cytokines (especially in IL-10 production (Supplementary Fig.1) ".

Response:

Appreciate for your valuable comment. In this revision, we have changed the sentence in line 131 as suggested:" Given the importance of Treg in secreting immunosuppressive cytokines and inhibiting the activation of both innate and adaptive immune cells, the statistically significant elevated levels of Treg may contribute to the suppression immune response observed in LDs."

Reviewer #2:

1. The authors have addressed my comments. The paper is now suitable for publication. It is an important contribution.

Response:

Thank you for your comments.